# NETWORK ARCHITECTURE SEARCH FOR DOMAIN ADAPTATION

## ABSTRACT

Deep networks have been used to learn transferable representations for domain adaptation. Existing deep domain adaptation methods systematically employ popular hand-crafted networks designed specifically for image-classification tasks, leading to sub-optimal domain adaptation performance. In this paper, we present Neural Architecture Search for Domain Adaptation (NASDA), a principle framework that leverages differentiable neural architecture search to derive the optimal network architecture for domain adaptation task. NASDA is designed with two novel training strategies: neural architecture search with multi-kernel Maximum Mean Discrepancy to derive the optimal architecture, and adversarial training between a feature generator and a batch of classifiers to consolidate the feature generator. We demonstrate experimentally that NASDA leads to state-of-the-art performance on several domain adaptation benchmarks.

## 1 INTRODUCTION

Supervised machine learning models ($\Phi$) aim to minimize the empirical test error ($\epsilon(\Phi(\mathbf{x}), \mathbf{y})$) by optimizing $\Phi$ on training data ($\mathbf{x}$) and ground truth labels ($\mathbf{y}$), assuming that the training and testing data are sampled *i.i.d* from the same distribution. While in practical, the training and testing data are typically collected from related domains under different distributions, a phenomenon known as domain shift (or domain discrepancy) (Quionero-Candela et al., 2009). To avoid the cost of annotating each new test data, Unsupervised Domain Adaptation (UDA) tackles domain shift by transferring the knowledge learned from a rich-labeled source domain ($P(\mathbf{x}^s, \mathbf{y}^s)$) to the unlabeled target domain ($Q(\mathbf{x}^t)$). Recently unsupervised domain adaptation research has achieved significant progress with techniques like discrepancy alignment (Long et al., 2017; Tzeng et al., 2014; Ghifary et al., 2014; Peng & Saenko, 2018; Long et al., 2015; Sun & Saenko, 2016), adversarial alignment (Xu et al., 2019a; Liu & Tuzel, 2016; Tzeng et al., 2017; Liu et al., 2018a; Ganin & Lempitsky, 2015; Saito et al., 2018; Long et al., 2018), and reconstruction-based alignment (Yi et al., 2017; Zhu et al., 2017; Hoffman et al., 2018; Kim et al., 2017). While such models typically learn feature mapping from one domain ($\Phi(\mathbf{x}^s)$) to another ($\Phi(\mathbf{x}^t)$) or derive a joint representation across domains ($\Phi(\mathbf{x}^s) \otimes \Phi(\mathbf{x}^t)$), the developed models have limited capacities in deriving an optimal neural architecture specific for domain transfer.

To advance network designs, neural architecture search (NAS) automates the net architecture engineering process by reinforcement supervision (Zoph & Le, 2017) or through neuro-evlolution (Real et al., 2019a). Conventional NAS models aim to derive neural architecture $\alpha$ along with the network parameters $w$, by solving a bilevel optimization problem (Anandalingam & Friesz, 1992): $\Phi_{\alpha,w} = \arg\min_\alpha \mathcal{L}_{val}(w^*(\alpha), \alpha)$ s.t. $w^*(\alpha) = \text{argmin}_w \mathcal{L}_{train}(w, \alpha)$, where $\mathcal{L}_{train}$ and $\mathcal{L}_{val}$ indicate the training and validation loss, respectively. While recent works demonstrate competitive performance on tasks such as image classification (Zoph et al., 2018; Liu et al., 2018c;b; Real et al., 2019b) and object detection (Zoph & Le, 2017), designs of existing NAS algorithms typically assume that the training and testing domain are sampled from the same distribution, neglecting the scenario where two data domains or multiple feature distributions are of interest.

To efficiently devise a neural architecture across different data domains, we propose a novel learning task called Neural Architecture Search for Domain Adaptation (NASDA). The ultimate goal of NASDA is to minimize the validation loss of the target domain ($\mathcal{L}_{val}^t$). We postulate that a solution to NASDA should not only minimize validation loss of the source domain ($\mathcal{L}_{val}^s$), but should also

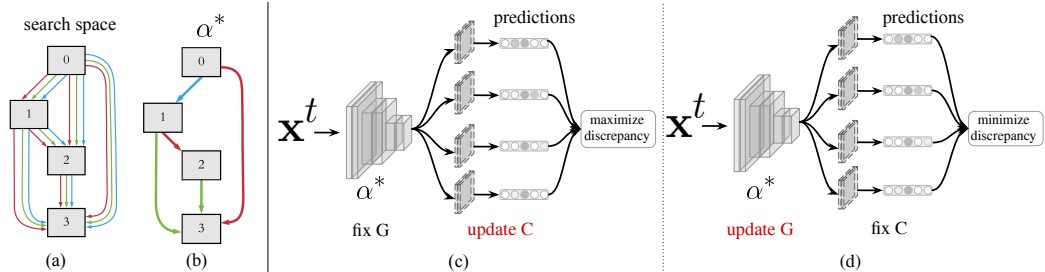

Figure 1: An overview of NASDA: (a) Continuous relaxation of the research space by placing a mixture of the candidate operations on each edge. (b) Inducing the final architecture by joint optimization of the neural architecture parameters $\alpha$ and network weights $w$, supervised by minimizing the validation loss on the source domain and reducing the domain discrepancy. (c)(d) Adversarial training of the derive feature generator $G$ and classifiers $C$.

reduce the domain gap between the source and target. To this end, we propose a new NAS learning schema:

$$\Phi_{\alpha,w} = \operatorname{argmin}_\alpha \mathcal{L}^s_{val}(w^*(\alpha), \alpha) + \operatorname{disc}(\Phi^*(\mathbf{x}^s), \Phi^*(\mathbf{x}^t)) \qquad (1)$$

$$\text{s.t.} \quad w^*(\alpha) = \operatorname{argmin}_w \ \mathcal{L}^s_{train}(w, \alpha) \qquad (2)$$

where $\Phi^* = \Phi_{\alpha, w^*(\alpha)}$, and $\operatorname{disc}(\Phi^*(\mathbf{x}^s), \Phi^*(\mathbf{x}^t))$ denotes the domain discrepancy between the source and target. Note that in unsupervised domain adaptation, $\mathcal{L}^t_{train}$ and $\mathcal{L}^t_{val}$ cannot be computed directly due to the lack of label in the target domain.

Inspired by the past works in NAS and unsupervised domain adaptation, we propose in this paper an instantiated NASDA model, which comprises of two training phases, as shown in Figure 1. The first is the neural architecture searching phase, aiming to derive an optimal neural architecture ($\alpha^*$), following the learning schema of Equation 1,2. Inspired by Differentiable ARchiTecture Search (DARTS) (Liu et al., 2019a), we relax the search space to be continuous so that $\alpha$ can be optimized with respect to $\mathcal{L}^s_{val}$ and $\operatorname{disc}(\Phi(\mathbf{x}^s), \Phi(\mathbf{x}^t))$ by gradient descent. Specifically, we enhance the feature transferability by embedding the hidden representations of the task-specific layers to a reproducing kernel Hilbert space where the mean embeddings can be explicitly matched by minimizing $\operatorname{disc}(\Phi(\mathbf{x}^s), \Phi(\mathbf{x}^t))$. We use multi-kernel Maximum Mean Discrepancy (MK-MMD) (Gretton et al., 2007) to evaluate the domain discrepancy.

The second training phase aims to learn a good feature generator with task-specific loss, based on the derived $\alpha^*$ from the first phase. To establish this goal, we use the derived deep neural network ($\Phi_{\alpha^*}$) as the feature generator ($G$) and devise an adversarial training process between $G$ and a batch of classifiers $C$. The high-level intuition is to first diversify $C$ in the training process, and train $G$ to generate features such that the diversified $C$ can have similar outputs. The training process is similar to Maximum Classifier Discrepancy framework (MCD) (Saito et al., 2018) except that we extend the dual-classifier in MCD to an ensembling of multiple classifiers. Experiments on standard UDA benchmarks demonstrate the effectiveness of our derived NASDA model in achieving significant improvements over state-of-the-art methods.

Our contributions of this paper are highlighted as follows:

- We formulate a novel dual-objective task of Neural Architecture Search for Domain Adaptation (NASDA), which optimize neural architecture for unsupervised domain adaptation, concerning both source performance objective and transfer learning objective.

- We propose an instantiated NASDA model that comprises two training stages, aiming to derive optimal architecture parameters $\alpha^*$ and feature extractor $G$, respectively. We are the first to show the effectiveness of MK-MMD in NAS process specified for domain adaptation.

- Extensive experiments on multiple cross-domain recognition tasks demonstrate that NASDA achieves significant improvements over traditional unsupervised domain adaptation models as well as state-of-the-art NAS-based methods.

## 2 RELATED WORK

Deep convolutional neural network has been dominating image recognition task. In recent years, many handcrafted architectures have been proposed, including VGG (Simonyan & Zisserman, 2014), ResNet (He et al., 2016), Inception (Szegedy et al., 2015), etc., all of which verifies the importance of human expertise in network design. Our work bridges domain adaptation and the emerging field of neural architecture search (NAS), a process of automating architecture engineering technique.

**Neural Architecture Search** Neural Architecture Search has become the mainstream approach to discover efficient and powerful network structures (Zoph & Le, 2017; Zoph et al., 2018). The automatically searched architectures have achieved highly competitive performance in tasks such as image classification (Liu et al., 2018c;b), object detection (Zoph et al., 2018), and semantic segmentation (Chen et al., 2018). Reinforce learning based NAS methods (Zoph & Le, 2017; Tan et al., 2019; Tan & Le, 2019) are usually computational intensive, thus hampering its usage with limited computational budget. To accelerate the search procedure, many techniques has been proposed and they mainly follow four directions: (**1**) estimating the actual performance with *lower fidelities*. Such lower fidelities include shorter training times (Zoph et al., 2018; Zela et al., 2018), training on a subset of the data (Klein et al., 2017), or on lower-resolution images. (**2**) estimating the performance based on the *learning curve extrapolation*. Domhan et al. (2015) propose to extrapolate initial learning curves and terminate those predicted to perform poorly. (**3**) *initializing* the novel architectures based on other well-trained architectures. Wei et al. (2016) introduce network morphisms to modify an architecture without changing the network objects, resulting in methods that only require a few GPU days (Elsken et al., 2017; Cai et al., 2018a; Jin et al., 2019; Cai et al., 2018b). (**4**) *one-shot* architecture search. One-shot NAS treats all architectures as different subgraphs of a supergraph and shares weights between architectures that have edges of this supergraph in common (Saxena & Verbeek, 2016; Liu et al., 2019b; Bender, 2018). DARTS (Liu et al., 2019a) places a mixture of candidate operations on each edge of the one-shot model and optimizes the weights of the candidate operations with a continuous relaxation of the search space. Inspired by DARTS (Liu et al., 2019a), our model employs differentiable architecture search to derive the optimal feature extractor for unsupervised domain adaptation.

**Domain Adaptation** Unsupervised domain adaptation (UDA) aims to transfer the knowledge learned from one or more labeled source domains to an unlabeled target domain. Various methods have been proposed, including discrepancy-based UDA approaches (Long et al., 2017; Tzeng et al., 2014; Ghifary et al., 2014; Peng & Saenko, 2018), adversary-based approaches (Liu & Tuzel, 2016; Tzeng et al., 2017; Liu et al., 2018a), and reconstruction-based approaches (Yi et al., 2017; Zhu et al., 2017; Hoffman et al., 2018; Kim et al., 2017). These models are typically designed to tackle single source to single target adaptation. Compared with single source adaptation, multi-source domain adaptation (MSDA) assumes that training data are collected from multiple sources. Originating from the theoretical analysis in (Ben-David et al., 2010; Mansour et al., 2009; Crammer et al., 2008), MSDA has been applied to many practical applications (Xu et al., 2018; Duan et al., 2012; Peng et al., 2019). Specifically, Ben-David et al. (2010) introduce an $\mathcal{H}\Delta\mathcal{H}$-divergence between the weighted combination of source domains and a target domain. These models are developed using the existing hand-crafted network architecture. This property limits the capacity and versatility of domain adaptation as the backbones to extract the features are fixed. In contrast, we tackle the UDA from a different perspective, not yet considered in the UDA literature. We propose a novel dual-objective model of NASDA, which optimize neural architecture for unsupervised domain adaptation. We are the first to show the effectiveness of MK-MMD in NAS process which is designed specifically for domain adaptation.

## 3 NEURAL ARCHITECTURE SEARCH FOR DOMAIN ADAPTATION

In unsupervised domain adaptation, we are given a source domain $\mathcal{D}_s = \{(\mathbf{x}_i^s, \mathbf{y}_i^s)\}_{i=1}^{n_s}$ of $n_s$ labeled examples and a target domain $\mathcal{D}_t = \{\mathbf{x}_j^t\}_{j=1}^{n_t}$ of $n_t$ unlabeled examples. The source domain and target domain are sampled from joint distributions $P(\mathbf{x}^s, \mathbf{y}^s)$ and $Q(\mathbf{x}^t, \mathbf{y}^t)$, respectively. The goal of this paper is to leverage NAS to derive a deep network $G : \mathbf{x} \mapsto \mathbf{y}$, which is optimal for reducing the shifts in data distributions across domains, such that the target risk $\epsilon_t(G) = \mathbb{E}_{(\mathbf{x}^t, \mathbf{y}^t) \sim Q}[G(\mathbf{x}^t) \neq \mathbf{y}^t]$ is minimized. We will start by introducing some preliminary background in Section 3.1. We then describe how to incorporate the MK-MMD into the neural architecture searching framework in

Section 3.2. Finally, we introduce the adversarial training between our derived deep network and a batch of classifiers in Section 3.3. An overview of our model can be seen in Algorithm 1.

## 3.1 PRELIMINARY: DARTS

In this work, we leverage DARTS (Liu et al., 2019a) as our baseline framework. Our goal is to search for a robust cell and apply it to a network that is optimal to achieve domain alignment between $\mathcal{D}_s$ and $\mathcal{D}_t$. Following Zoph et al. (2018), we search for a computation cell as the building block of the final architecture. The final convolutional network for domain adaptation can be stacked from the learned cell. A cell is defined as a directed acyclic graph (DAG) of $L$ nodes, $\{x^i\}_{i=1}^N$, where each node $x^{(i)}$ is a latent representation and each directed edge $\mathrm{e}^{(i,j)}$ is associated with some operation $o^{(i,j)}$ that transforms $x^{(i)}$. DARTS (Liu et al., 2019a) assumes that cells contain two input nodes and a single output node. To make the search space continuous, DARTS relaxes the categorical choice of a particular operation to a softmax over all possible operations and is thus formulated as:

$$\bar{o}^{(i,j)}(x) = \sum_{o \in \mathcal{O}} \frac{\exp(\alpha_o^{(i,j)})}{\sum_{o' \in \mathcal{O}} \exp(\alpha_{o'}^{(i,j)})} o(x) \tag{3}$$

where $\mathcal{O}$ denotes the set of candidate operations and $i < j$ so that *skip-connect* can be applied. An intermediate node can be represented as $x_j = \sum_{i<j} o^{(i,j)}(x_i)$. The task of architecture search then reduces to learning a set of continuous variables $\alpha = \{\alpha^{(i,j)}\}$. At the end of search, a discrete architecture can be obtained by replacing each mixed operation $\bar{o}^{(i,j)}$ with the most likely operation, i.e., $o^{*(i,j)} = \mathrm{argmax}_{o \in \mathcal{O}} \ \alpha_o^{(i,j)}$ and $\alpha^* = \{o^{*(i,j)}\}$.

## 3.2 SEARCHING NEURAL ARCHITECTURE

Denote by $\mathcal{L}_{train}$ and $\mathcal{L}_{val}$ the training loss and validation loss, respectively. Conventional neural architecture search models aim to derive $\Phi_{\alpha,w}$ by solving a bilevel optimization problem (Anandalingam & Friesz, 1992): $\Phi_{\alpha,w} = \arg\min_\alpha \mathcal{L}_{val}(w^*(\alpha), \alpha)$ s.t. $w^*(\alpha) = \mathrm{argmin}_w \mathcal{L}_{train}(w, \alpha)$. While recent work (Zoph et al., 2018; Liu et al., 2018c) have show promising performance on tasks such as image classification and object detection, the existing models assume that the training data and testing data are sampled from the same distributions. Our goal is to jointly learn the architecture $\alpha$ and the weights $w$ within all the mixed operations (e.g. weights of the convolution filters) so that the derived model $\Phi_{w^*,\alpha^*}$ can transfer knowledge from $\mathcal{D}_s$ to $\mathcal{D}_t$ with some simple domain adapation guidance. Initialized by Equation 1, we leverage multi-kernel Maximum Mean Discrepancy (Gretton et al., 2007) to evaluate $\mathrm{disc}(\Phi^*(\mathbf{x}^s), \Phi^*(\mathbf{x}^t))$.

**MK-MMD** Denote by $\mathcal{H}_k$ be the Reproducing Kernel Hilbert Space (RKHS) endowed with a characteristic kernel $k$. The *mean embedding* of distribution $p$ in $\mathcal{H}_k$ is a unique element $\mu_k(P)$ such that $\mathbf{E}_{\mathbf{x} \sim P} f(\mathbf{x}) = \langle f(\mathbf{x}), \mu_k(P) \rangle_{\mathcal{H}_k}$ for all $f \in \mathcal{H}_k$. The MK-MMD $d_k(P,Q)$ between probability distributions $P$ and $Q$ is defined as the RKHS distance between the mean embeddings of $P$ and $Q$. The squared formulation of MK-MMD is defined as

$$d_k^2(P,Q) \triangleq \left\| \mathbf{E}_P \left[ \Phi_\alpha(\mathbf{x}^s) \right] - \mathbf{E}_Q \left[ \Phi_\alpha(\mathbf{x}^t) \right] \right\|_{\mathcal{H}_k}^2. \tag{4}$$

In this paper, we consider the case of combining Gaussian kernels with injective functions $f_\Phi$, where $k(x, x') = \exp(-\|f_\Phi(x) - f_\Phi(x)'\|^2)$. Inspired by Long et al. (2015), the characteristic kernel associated with the feature map $\Phi$, $k(\mathbf{x}^s, \mathbf{x}^t) = \langle \Phi(\mathbf{x}^s), \Phi(\mathbf{x}^t) \rangle$, is defined as the convex combination of $n$ positive semidefinite kernels $\{k_u\}$,

$$\mathcal{K} \triangleq \left\{ k = \sum_{u=1}^n \beta_u k_u : \sum_{u=1}^n \beta_u = 1, \beta_u \geqslant 0, \forall u \right\}, \tag{5}$$

where the constraints on $\{\beta_u\}$ are imposed to guarantee that the $k$ is characteristic. In practice we use finite samples from distributions to estimate MMD distance. Given $\mathbf{X}^s = \{\mathbf{x}_1^s, \cdots, \mathbf{x}_m^s\} \sim P$ and $\mathbf{X}^t = \{\mathbf{x}_1^t, \cdots, \mathbf{x}_m^t\} \sim Q$, one estimator of $d_k^2(P,Q)$ is

$$\hat{d}_k^2(P,Q) = \frac{1}{\binom{m}{2}} \sum_{i \neq i'} k(\mathbf{x^s}_i, \mathbf{x^s}_i') - \frac{2}{\binom{m}{2}} \sum_{i \neq j} k(\mathbf{x}_i^s, \mathbf{x}_j^t) + \frac{1}{\binom{m}{2}} \sum_{j \neq j'} k(\mathbf{x^t}_j, \mathbf{x^t}_j'). \tag{6}$$

---

**Algorithm 1** Neural Architecture Search for Domain Adaptation

---

**Phase I: Searching Neural Architecture**

 1: Create a mixed operation $o^{(i,j)}$ parametrized by $\alpha^{(i,j)}$ for each edge $(i,j)$
 2: **while** not converged **do**
 3:      Update architecture $\alpha$ by $\frac{\partial}{\partial \alpha} \mathcal{L}^s_{val}\left(w - \xi \frac{\partial}{\partial w} \mathcal{L}^s_{train}(w, \alpha), \alpha\right) + \lambda \frac{\partial}{\partial \alpha}\left(\hat{d}^2_k\left(\Phi(\mathbf{x}^s), \Phi(\mathbf{x}^t)\right)\right)$
 4:      Update weights $w$ by descending $\frac{\partial}{\partial w} \mathcal{L}^s_{train}(w, \alpha)$
 5: **end while**
 6: Derive the final architecture based on the learned $\alpha^*$.

**Phase II: Adversarial Training for Domain Adaptation**

 1: Stack feature generator $G$ based on $\alpha^*$, initialize classifiers $C$
 2: **while** not converged **do**
 3:      Step one: Train $G$ and $C$ with $\mathcal{L}^s(\mathbf{x}^s, \mathbf{y}^s) = -\mathbb{E}_{(\mathbf{x}^s,\mathbf{y}^s)\sim \mathcal{D}^s} \sum_{k=1}^K \mathbf{1}_{[k=\mathbf{y}^s]} \log p(\mathbf{y}^s|\mathbf{x}^s)$
 4:      Step two: Fix G, train C with loss: $\mathcal{L}^s(\mathbf{x}^s, \mathbf{y}^s) - \mathcal{L}_{\text{adv}}(\mathbf{x}^t)(Eq.\ 13)$
 5:      Step three: Fix C, train G with loss: $\mathcal{L}_{\text{adv}}(\mathbf{x}^t)$
 6: **end while**

---

The merit of multi-kernel MMD lies in its differentiability such that it can be easily incorporated into the deep network. However, the computation of the $\hat{d}^2_k(P, Q)$ incurs a complexity of $O(m^2)$, which is undesirable in the differentiable architecture search framework. In this paper, we use the unbiased estimation of MK-MMD (Gretton et al., 2012) which can be computed with linear complexity.

**NAS for Domain Adaptation** Denote by $\mathcal{L}^s_{train}$ and $\mathcal{L}^s_{val}$ the training loss and validation loss on the source domain, respectively. Both losses are affected by the architecture $\alpha$ as well as by the weights $w$ in the network. The goal for NASDA is to find $\alpha^*$ that minimizes the validation loss $\mathcal{L}^t_{val}(w^*, \alpha^*)$ on the target domain, where the weights $w^*$ associated with the architecture are obtained by minimizing the training loss $w^* = \operatorname{argmin}_w \mathcal{L}^s_{train}(w, \alpha^*)$. Due to the lack of labels in the target domain, it is prohibitive to compute $\mathcal{L}^t_{val}$ directly, hampering the assumption of previous gradient-based NAS algorithms (Liu et al., 2019a; Chen et al., 2019). Instead, we derive $\alpha^*$ by minimizing the validation loss $\mathcal{L}^s_{val}(w^*, \alpha^*)$ on the source domain plus the domain discrepancy, $\operatorname{disc}(\Phi(\mathbf{x}^s), \Phi(\mathbf{x}^t))$, as shown in Equation 1.

Inspired by the gradient-based hyperparameter optimization (Franceschi et al., 2018; Pedregosa, 2016; Maclaurin et al., 2015), we set the architecture parameters $\alpha$ as a special type of hyperparameter. This implies a bilevel optimization problem (Anandalingam & Friesz, 1992) with $\alpha$ as the upper-level variable and $w$ as the lower-level variable. In practice, we utilize the MK-MMD to evaluate the domain discrepancy. The optimization can be summarized as follows:

$$\Phi_{\alpha,w} = \operatorname{argmin}_\alpha \left(\mathcal{L}^s_{val}(w^*(\alpha), \alpha) + \lambda \hat{d}^2_k\left(\Phi(\mathbf{x}^s), \Phi(\mathbf{x}^t)\right)\right) \qquad (7)$$

$$\text{s.t. } w^*(\alpha) = \operatorname{argmin}_w \mathcal{L}^s_{train}(w, \alpha) \qquad (8)$$

where $\lambda$ is the trade-off hyperparameter between the source validation loss and the MK-MMD loss.

**Approximate Architecture Search** Equation 7,8 imply that directly optimizing the architecture gradient is prohibitive due to the expensive inner optimization. Inspired by DARTS (Liu et al., 2019a), we approximate $w^*(\alpha)$ by adapting $w$ using only a single training step, without solving the optimization in Equation 8 by training until convergence. This idea has been adopted and proven to be effective in meta-learning for model transfer (Finn et al., 2017), gradient-based hyperparameter tuning (Luketina et al., 2016) and unrolled generative adversarial networks. We therefore propose a simple approximation scheme as follows:

$$\frac{\partial}{\partial \alpha}\left(\mathcal{L}^s_{val}(w^*(\alpha), \alpha) + \lambda \hat{d}^2_k\left(\Phi(\mathbf{x}^s), \Phi(\mathbf{x}^t)\right)\right)$$
$$\approx \frac{\partial}{\partial \alpha} \mathcal{L}^s_{val}\left(w - \xi \frac{\partial}{\partial w} \mathcal{L}^s_{train}(w, \alpha), \alpha\right) + \lambda \frac{\partial}{\partial \alpha}\left(\hat{d}^2_k\left(\Phi(\mathbf{x}^s), \Phi(\mathbf{x}^t)\right)\right) \qquad (9)$$

where $w - \xi \frac{\partial}{\partial w} \mathcal{L}^s_{train}(w, \alpha)$ denotes weight for one-step forward model and $\xi$ is the learning rate for a step of inner optimization. Note Equation 9 reduces to $\nabla_\alpha \mathcal{L}_{val}(w, \alpha)$ if $w$ is already a local optimum for the inner optimization and thus $\nabla_w \mathcal{L}_{train}(w, \alpha) = 0$.

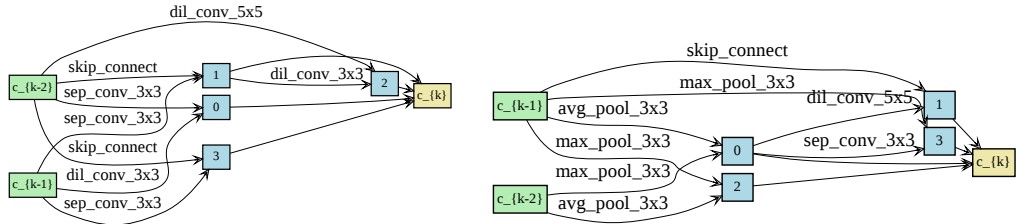

(a) Normal cells (left) and Reduce cells (right) for STL→CIFAR10

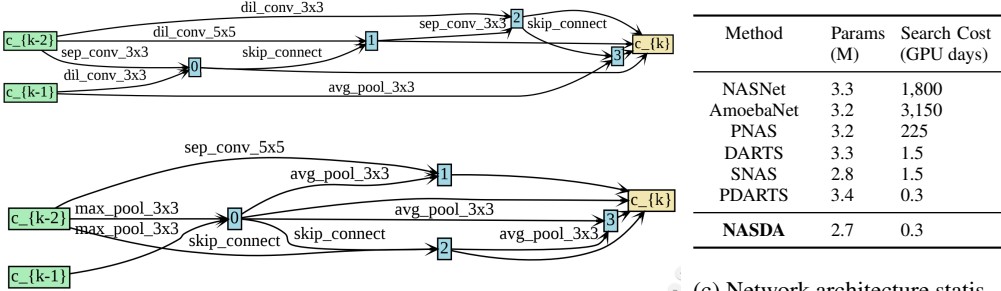

(c) Network architecture statis-

(b) Normal cells (upper) and Reduce cells (lower) for MNIST→USPS  tics comparison

| Method | Params (M) | Search Cost (GPU days) |
|---|---|---|
| NASNet | 3.3 | 1,800 |
| AmoebaNet | 3.2 | 3,150 |
| PNAS | 3.2 | 225 |
| DARTS | 3.3 | 1.5 |
| SNAS | 2.8 | 1.5 |
| PDARTS | 3.4 | 0.3 |
| **NASDA** | 2.7 | 0.3 |

Figure 2: (a) Neural architecture for STL→CIFAR10 task. (b) Neural architecture for MNIST→USPS results. (c) Comparison between our NASDA model and state-of-the-art NAS models.

The second term of Equation 9 can be computed directly with some forward and backward passes. For the first term, applying chain rule to the approximate architecture gradient yields

$$\frac{\partial}{\partial \alpha} \mathcal{L}^s_{val}(w', \alpha) - \xi \big( \frac{\partial^2}{\partial \alpha \partial w} \mathcal{L}^s_{train}(w, \alpha) \frac{\partial}{\partial w'} \mathcal{L}^s_{val}(w', \alpha) \big) \tag{10}$$

where $w' = w - \xi \frac{\partial}{\partial w} \mathcal{L}_{train}(w, \alpha)$. The expression above contains an expensive matrix-vector product in its second term. We leverage the central difference approximation to reduce the computation complexity. Specifically, let $\eta$ be a small scalar and $w^{\pm} = w \pm \eta \frac{\partial}{\partial w'} \mathcal{L}^s_{val}(w', \alpha)$. Then:

$$\frac{\partial^2}{\partial \alpha \partial w} \mathcal{L}^s_{train}(w, \alpha) \frac{\partial}{\partial w'} \mathcal{L}^s_{val}(w', \alpha) \approx \frac{\frac{\partial}{\partial \alpha} \mathcal{L}_{train}(w^+, \alpha) - \frac{\partial}{\partial \alpha} \mathcal{L}_{train}(w^-, \alpha)}{2\eta} \tag{11}$$

Evaluating the central difference only requires two forward passes for the weights and two backward passes for $\alpha$, reducing the complexity from quadratic to linear.

### 3.3 ADVERSARIAL TRAINING FOR DOMAIN ADAPTATION

By neural architecture searching from Section 3.2, we have derived the optimal cell structure ($\alpha^*$) for domain adaptation. We then stack the cells to derive our feature generator $G$. In this section, we describe how do we consolidate $G$ by an adversarial training of $G$ and the classifiers $C$. Assume $C$ includes $N$ independent classifiers $\{C^{(i)}\}_{i=1}^N$ and denote $p_i(\mathbf{y}|\mathbf{x})$ as the $K$-way propabilistic outputs of $C^{(i)}$, where $K$ is the category number.

The high-level intuition is to consolidate the feature generator $G$ such that it can make the diversified $C$ generate similar outputs. To this end, our training process include three steps: (1) train $G$ and $C$ on $\mathcal{D}_s$ to obtain task-specific features, (2) fix $G$ and train $C$ to make $\{C^{(i)}\}_{i=1}^N$ have diversified output, (3) fix $C$ and train $G$ to minimize the output discrepancy between $C$. Related techniques have been used in Saito et al. (2018); Kumar et al. (2018).

First, we train both $G$ and $C$ to classify the source samples correctly with cross-entropy loss. This step is crucial as it enables $G$ and $C$ to extract the task-specific features. The training objective is $\min_{G,C} \mathcal{L}^s(\mathbf{x}^s, \mathbf{y}^s)$ and the loss function is defined as follows:

$$\mathcal{L}^s(\mathbf{x}^s, \mathbf{y}^s) = -\mathbb{E}_{(\mathbf{x}^s, \mathbf{y}^s) \sim \mathcal{D}^s} \sum_{k=1}^K \mathbf{1}_{[k=\mathbf{y}^s]} \log p(\mathbf{y}^s|\mathbf{x}^s) \tag{12}$$

In the second step, we are aiming to diversify $C$. To establish this goal, we fix $G$ and train $C$ to increase the discrepancy of $C$'s output. To avoid mode collapse (*e.g.* $C^{(1)}$ outputs all zeros and $C^{(2)}$ output all ones), we add $\mathcal{L}^s(\mathbf{x}^s, \mathbf{y}^s)$ as a regularizer in the training process. The high-level intuition is that we do not expect $C$ to forget the information learned in the first step in the training process. The training objective is $\min_C \mathcal{L}^s(\mathbf{x}^s, \mathbf{y}^s) - \mathcal{L}_{\mathrm{adv}}(\mathbf{x}^t)$, where the adversarial loss is defined as:

$$\mathcal{L}_{\mathrm{adv}}(\mathbf{x}^t) = \mathbb{E}_{\mathbf{x}^t \sim \mathcal{D}^t} \sum_{i=1}^{N-1} \sum_{j=i+1}^{N} \|(p_i(\mathbf{y}|\mathbf{x}^t) - p_j(\mathbf{y}|\mathbf{x}^t)\|_1 \tag{13}$$

In the last step, we are trying to consolidate the feature generator $G$ by training $G$ to extract generalizable representations such that the discrepancy of $C$'s output is minimized. To achieve this goal, we fix the diversified classifiers $C$ and train $G$ with the adversarial loss (defined in Equation 13). The training objective is $\min_G \mathcal{L}_{\mathrm{adv}}(\mathbf{x}_t)$.

## 4 EXPERIMENTS

We compare the proposed NASDA model with many state-of-the-art UDA baselines on multiple benchmarks. In the main paper, we only report major results; more details are provided in the supplementary material. All of our experiments are implemented in the PyTorch platform.

In the architecture search phase, we use $\lambda=1$ for all the searching experiments. We leverage the ReLU-Conv-BN order for convolutional operations, and each separable convolution is always applied twice. Our search space $\mathcal{O}$ includes the following operations: $3 \times 3$ and $5 \times 5$ separable convolutions, $3 \times 3$ and $5 \times 5$ dilated separable convolutions, $3 \times 3$ max pooling, identity, and *zero*. Our convolutional cell consists of $N = 7$ nodes. Cells located at the $\frac{1}{3}$ and $\frac{2}{3}$ of the total depth of the network are reduction cells. The architecture encoding therefore is

| Method | STL → CIFAR10 |
|---|---|
| DANN | 56.9 |
| MCD | 69.2 |
| DWT | 71.2 |
| SE | 74.2 |
| G2A | 72.8 |
| VADA | 73.5 |
| DIRT-T | 75.3 |
| NASNet+*Phase II* | 67.3 |
| AmoebaNet+*Phase II* | 67.0 |
| DARTS+*Phase II* | 68.8 |
| PDARTS+*Phase II* | 66.0 |
| **NASDA** | **76.8** |

Table 1: Accuracy (%) STL→CIFAR-10

$(\alpha_{normal}, \alpha_{reduce})$, where $\alpha_{normal}$ is shared by all the normal cells and $\alpha_{reduce}$ is shared by all the reduction cells.

### 4.1 SETUP

**Digits** We investigate three digits datasets: **MNIST**, **USPS**, and Street View House Numbers (**SVHN**). We adopt the evaluation protocol of CyCADA (Hoffman et al., 2018) with three transfer tasks: USPS to MNIST ($\mathbf{U} \to \mathbf{M}$), MNIST to USPS ($\mathbf{M} \to \mathbf{U}$), and SVHN to MNIST ($\mathbf{S} \to \mathbf{M}$). We train our model using the training sets: MNIST (60,000), USPS (7,291), standard SVHN train (73,257).

**STL→CIFAR10** Both CIFAR10 (Krizhevsky et al., 2009) and STL (Coates et al., 2011) are both 10-class image datasets. These two datasets contain nine overlapping classes. We remove the 'frog' class in CIFAR10 and the 'monkey' class in STL datasets as they have no equivalent in the other dataset, resulting in a 9-class problem. The STL images were down-scaled to $32 \times 32$ resolution to match that of CIFAR10.

**SYN SIGNS→GTSRB** We evaluated the adaptation from synthetic traffic sign dataset called SYN SIGNS (Moiseev et al., 2013) to real-world sign dataset called GTSRB (Stallkamp et al., 2011). These datasets contain 43 classes.

We compare our NASDA model with state-of-the-art DA methods: Deep Adaptation Network (**DAN**) (Long et al., 2015), Domain Adversarial Neural Network (**DANN**) (Ganin & Lempitsky, 2015), Domain Separation Network (**DSN**) (Bousmalis et al., 2016), Coupled Generative Adversarial Networks (**CoGAN**) (Liu & Tuzel, 2016), Maximum Classifier Discrepancy (**MCD**) (Saito et al., 2018), Generate to Adapt (**G2A**) (Sankaranarayanan et al., 2018), Stochastic Neighborhood Embedding (**d-SNE**) (Xu et al., 2019b), Associative Domain Adaptation (**ASSOC**) (Haeusser et al., 2017).

| Method | M → U | U → M | S → M | Avg(digits) | Method | SYN SIGNS → GTSRB |
|---|---|---|---|---|---|---|
| DAN | 81.1 | - | 71.1 | 76.1 | DAN | 91.1 |
| DANN | 85.1 | 73.0 | 71.1 | 76.4 | DANN | 88.7 |
| DSN | 91.3* | - | 82.7 | - | DSN | 93.1 |
| CoGAN | 91.2* | 89.1 | - | - | CORAL | 86.9 |
| MCD | 94.2 | 94.1 | 96.2 | 94.8 | MCD | 94.4 |
| G2A | 95.0 | 90.8 | 92.4 | 92.7 | ASSOC | 82.8 |
| SBADA-GAN | 95.3 | 97.6 | 76.1 | 89.7 | SRDA-RAN | 93.6 |
| d-SNE | **99.0** | 98.7 | 96.5 | 98.1 | DADRL | 94.6 |
| **NASDA** | 98.0 | **98.7** | **98.6** | **98.4** | **NASDA** | **96.7** |

Table 2: Accuracy (%) on Digits and Traffic Signs for unsupervised domain adaptation.

## 4.2 EMPIRICAL RESULTS

**Neural Architecture Search Results** We show the neural architecture search results in Figure 2. We can observe that our model contains more "avg_pool" and "5x5_conv" layer than other NAS model. This will make our model more generic as the average pooling method smooths out the image and hence the model is not congested with sharp features and domain-specific features. We also show that our NASDA model contains less parameters and takes less time to converge compared with state-of-the-art NAS architectures. Another interesting finding is that our NASDA contains more sequential connections in both Normal and Reduce cells when trained on MNIST→USPS.

**Unsupervised Domain Adaptation Results** The UDA results for Digits and SYN SIGNS→GTSRB are reported in Table 2, with results of baselines directly reported from the original papers if the protocol is the same (numbers with * indicates training on partial data). The NASDA model achieves a **98.4%** average accuracy for Digits dataset, outperforming other baselines. For SYN SIGNS→GTSRB task, our model gets comparable results with state-of-the-art baselines. The results demonstrate the effectiveness of our NASDA model on small images.

The UDA results on the STL→CIFAR10 recognition task are reported in Table 1. Our model achieves a performance of **76.8%**, outperforming all the baselines. To compare our search neural architecture with previous NAS models, we replace the neural architecture in $G$ with other NAS models. Other training settings in the second phase are identical to our model. As such, we derive **NASNet**+*Phase II* (Zoph et al., 2018), **AmoebaNet**+*Phase II* (Shah et al., 2018) , **DARTS**+*Phase II* (Liu et al., 2019a), and **PDARTS**+*Phase II* (Chen et al., 2019) models. The results in Table 1 demonstrate that our model outperform other NAS based model by a large margin, which shows the effectiveness of our model in unsupervised domain adaptation. Specifically, we set **PDARTS**+*Phase II* as an ablation study to demonstrate the effectiveness of our task-specific design in learning domain-adaption aware features.

**Analysis** To dive deeper into the training process of our NASDA model, we plot the T-SNE embedding of the weights of $C$ in USPS→MNIST in Figure 3. This is achieved by recording the weights of all the classifiers at each epoch. The black dot indicates epoch zero, which is the common starting point. The color from light to dark corresponds to the epoch number from small to large. The T-SNE plots clearly show that the classifiers are diverged from each other, demonstrating the effectiveness of the second step of our NASDA training described in Section 3.2.

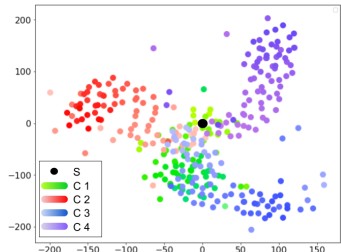

Figure 3: T-SNE for 4 classifiers

## 5 CONCLUSION

In this paper, we first formulate a novel dual-objective task of Neural Architecture Search for Domain Adaptation (NASDA) to invigorate the design of transfer-aware network architectures. Towards tackling the NASDA task, we have proposed a novel learning framework that leverages MK-MMD to guide the neural architecture search process. Instead of aligning the features from existing handcrafted backbones, our model directly searches for the optimal neural architecture specific for domain adaptation. Furthermore, we have introduced the ways to consolidate the feature generator, which is stacked from the searched architecture, in order to boost the UDA performance. Extensive empirical evaluations on UDA benchmarks have demonstrated the efficacy of the proposed model against several state-of-the-art domain adaptation algorithms.

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
