# OpenReview forum: "Network Architecture Search for Domain Adaptation"
_ICLR.cc/2021/Conference — Reject_

### Official Review · AnonReviewer3 · 2020-10-26
**A method combining NAS with DA but lack of enough technique novelty, comprehensive experiments, and ablation studies.**

**Rating:** 4
**Confidence:** 5

**Review:**

In the work, the authors aim at improving the transferability of domain adaptation models from the perspective of neural architecture search. It consists of two phases, in particular, the first phase searches a neural architecture for domain adaptation based on a famous differentiable NAS method named DARTs, and the second phase develops an adversarial training method for domain adaptation by extending MCD to a multiple classifiers version. The empirical study evaluates the performance on some UDA tasks and shows the effectiveness of the introduced method.

Pros:
1.	Tackling the problem of domain adaptation from the perspective of neural architecture search is interesting, and the authors successfully combine them together.

2.	They not only minimize the validation loss of the source domain but also reduce the domain gap between the source and target since the labels in the target domain are not available. Though somewhat incremental,  this design works in some NAS benchmarks. On the contrary, the extension of MCD to multiple classifiers is somewhat straightforward.

3.	This paper is well-written and easy to follow, and the empirical result on one small-scale benchmark named STL → CIFAR10 is impressive.

Concerns:
1.	As for the first phase, jointly minimizing the validation loss of the source domain and the MK-MMD based domain gap between the source and target, however, can NOT guarantee a small target error according to the analysis of [1], in which a simple counterexample is given in Figure 1 of [1]. My question is, by searching a differentiable neural architecture with the objective function defined in Equation 7, how can we guarantee it is suitable in the target domain and ensure a high target accuracy. The authors may want to search for a not so bad architecture first and then align the features across domains, however, the main purpose of this paper is the former but not the latter. Therefore, ablation studies are necessary to verify the effectiveness of the MK-MMD loss, such as comparing the architecture of the proposed method with that of DARTs. Note that, in my opinion, this comparison should be done without the second phase and different from the protocol of Table 1 to avoid the interference of the second phase.

2. The technique novelty of the first phase of combining DARTS with MK-MMD is somewhat incremental and limited. It seems that the first phase is just an instance of famous DARTS (proposed in the field of NAS) with the additional popular MK-MMD loss function which also has already been proposed in the field of domain adaptation. For me, neither new ideas in the field of DA nor new techniques in the field of NAS are seen.

3.	As for the second phase, the extension of MCD to multiple classifiers is straightforward and somewhat incremental. Meanwhile, I am still curious about the performance if we increase the classifier number from 2 to 4 or some larger. If we simply ensemble several classifiers without adversarial training, we have reasons to believe that we can achieve higher accuracy than the model with two classifiers since the effectiveness of ensemble learning has been verified in literature.

4.	As shown in Table 2, the results of other NAS models with the same phase II are extremely low, however, there is no much difference between the architecture of NASDA with that of other NAS models. Especially, the architecture of NASNet [2] is also designed to be transferable across datasets (from Cifar to Imagenet), I am quite curious why this kind of architectures except NASDA are so weak. Would you mind giving some convincing analysis or explanations? It would be useful to shed more light on the differences with other NAS models, it is still unclear why these differences make such a big difference in the results. Meanwhile, the author has better show some evidence to justify that NAS can find more domain-invariant features than the classical architecture including VGG, ResNet, DenseNet.

5.	Another key concern about the paper is the lack of rigorous experimentation to study the usefulness of the proposed method. State-of-the-art performances are more impressive and convincing when the benchmark adopts realistic settings. The typical domain adaptation benchmarks, such as Office-31, ImageCLEF-DA, VisDA, Office-Home, and DomainNet, as well as the newly proposed ImageNet-Sketch, have various category number and dataset scale. It would be better to conduct some experiments on these standard domain adaptation datasets and compare the proposed method with numerous baselines of these benchmarks.


[1] Han Zhao⋆, Remi Tachet des Combes†, Kun Zhang⋆, Geoffrey J. Gordon⋆,† Carnegie Mellon University⋆, Microsoft Research Montrea. On Learning Invariant Representation for Domain Adaptation. In ICML 2019.

[2] Barret Zoph, Vijay Vasudevan, Jonathon Shlens, and Quoc V Le. Learning transferable architectures for scalable image recognition. In Proceedings of the IEEE conference on computer vision and pattern recognition, pages 8697– 8710, 2018

---

### Official Review · AnonReviewer2 · 2020-10-26
**Interesting and unstudied topic but the method is of limited originality and not always well justified**

**Rating:** 4
**Confidence:** 4

**Review:**

This paper introduces an approach to search for the best network architecture for a domain adaptation task. This is achieved by following a differentiable architecture search strategy in which an additional loss function is included to account for the domain shift. Specifically, the loss function aims to minimize the discrepancy between feature representations from the two domains.

Strengths:
- To the best of my knowledge, this constitutes the first attempt at performing neural architecture search for domain adaptation
- The results showing that accounting for the DA task during NAS outperforms simply taking a NAS model and applying a DA strategy to it are encouraging.


Weaknesses:

Originality:
- While I like the general concept of designing a NAS method specifically for domain adaptation, the proposed method lacks originality. In essence, it complements a standard NAS approach with standard DA strategies. This is fine, and the results are encouraging, but this seems to be a small contribution for an ICLR paper.

Methodology:
- Something bothers me in the formulation of Eqs. (1)-(2). While the discrepancy term of course depends on the architecture, it also depends on the weights w. It is therefore not very intuitive to my why the discrepancy term does not appear in the inner minimization problem of Eq. (2).
- The other point that I find disturbing in the methodology is the fact that, during the second training stage, a different DA strategy, relying on adversarial training instead of MK-MMD, is used. Why not train the final model with MK-MMD? Alternatively, why not rely on adversarial training during the search phase? This unfortunately makes the overall method less principled.
- As mentioned by the authors, one cannot optimized the target validation loss, because the target data is unsupervised. However, it would be interesting to evaluate what happens using pseudo-labeling, which has proven effective in DA, to approximate the target validation loss.

Experiments:
- In the comparison with the state-of-the-art DA methods in Tables 1 and 2, what architectures do these methods rely on? Are the comparisons fair, in the sense that all methods use networks with similar capacity?
- The datasets used here are a bit outdated, and I recommend the authors to rely on newer benchmarks, e.g., VisDA or office-home.
- The set of baselines differs across the different experiments. For example, d-SNE, which got the second best results on the digits datasets, is not reported for traffic signs and STL->CIFAR; DIRT-T that got the second-best results in Table 1 is not reported in Table 2.

Clarity:
- The paper is reasonably easy to understand, but could benefit from thorough proofreading.

---

### Official Review · AnonReviewer4 · 2020-10-27
**A review for Network Architecture Search for Domain Adaptation**

**Rating:** 4
**Confidence:** 5

**Review:**

**Summarize what the paper claims to contribute.**

This work introduces a two-step procedure for unsupervised domain adaptation; (1) neural architecture search for domain adaptation (NASDA) based on DARTS for neural architecture search and MK-MMD for differentiable unsupervised domain adaptation (2) adversarial training with a batch of classifiers. Based on induced architecture from the first component, it demonstrates unsupervised domain adaptation task supported by the second component.

**List strong and weak points of the paper.**

*strengths*

It is the first paper to adopt neural architecture search on unsupervised domain adaptation. The proposed combination, which is DARTS and MK-MMD, discovers a relevant neural architecture for unsupervised domain adaptation as shown in the experiment section.

*weaknesses*

The novelty of both components is limited.
The novelty of the first component is not explained well in the paper. The first component seems to be a simple combination of DARTS with the additional loss function, MK-MMD. I can’t find any new modification on DARTS with MK-MMD in the paper. All descriptions on <section 3.2 searching neural architecture> are borrowed from the previous works, DARTS and MK-MMD.
The novelty of the second component is not related to neural architecture search for domain adaptation but multiple discriminators for generative adversarial networks. In this aspect, the ablation study on the second component is not concluded.

**Clearly state your recommendation (accept or reject) with one or two key reasons for this choice**

I give “Ok but not good enough - rejection (4)” to this paper. Although the proposed method deals with new and fancy stuff, the novelty of both proposed components is limited.

I believe that the author should give more evidence to support the claims. If the author wants to insist that the first component, adopting MK-MMD on DARTS, is a good option, authors should prove that NASDA is effective for searching transferrable architecture by (1) giving a theoretical proof or (2) showing good performance on various practical benchmarks for domain adaptation. For example on argument (1), NASDA gives a tighter lower bound compared to a naive DA counterpart. For example on argument (2), NASDA shows good performance on several benchmarks such as office-31, ImageCLEF, and VisDA or multi-source domain benchmark.

If the author wants to prove the effectiveness of the second component, the author should compare the proposed method to (1) single discriminator and (2) the other multiple discriminator schemes. But even this comparison is not related to neural architecture search.

**Provide additional feedback with the aim to improve the paper.**

*Inaccurate descriptions on the paper*

The description of [conventional NAS ~ training and validation loss, respectively] in the second paragraph of the introduction section is that of DARTS (Liu, et. al., 2019), not generic NAS. For example, the work (Zoph & Le, 2017) generates a neural architecture to maximize the expected accuracy of the generated architecture on the validation set.

---

### Official Review · AnonReviewer1 · 2020-10-28
**Interesting work, can improve with minor additions**

**Rating:** 6
**Confidence:** 3

**Review:**


This work devises a two step process for searching optimal models for unsupervised domain adaptation. The first step involves a modification of DARTS where a discrepancy term between features for source and target domain is added to the negative-reward. The obtained feature transformer is then re-trained with an adversarial objective in order to ensure that it performs well across multiple classifiers.

The work addresses an interesting problem of automatically finding suitable architectures that transfer to unlabelled datasets. The novelty is involved in the formulation of a protocol that combines a modified DARTS objective with a post-processing step to get a suitable feature generator. The experiment section is well detailed.

I have following concerns with respect to the current state of this work
- It would be useful to get insights on the number of classifiers used in Phase-2 for current experiments along with the influence of this hyperparameter on the overall performance.
- Is it possible to extend PDARTS with the MK-MMD regularisation term? Do authors have any insights in this direction?
- Establishing the independent effectiveness of Phase-2 and regularisation term will add value to this work. These would also serve as relevant baselines.
- If possible, could the authors detail the experimental setup wrt different NAS algorithms as it has been shown that often use of simple data augmentation techniques can lead to significant changes in performance.
- Similarly, what part of the network was used as the feature generator? Does the performance vary with the number of layers used for G? Since the introduction of MK-MDD term led to models with reduced parameters as per the table in Figure 2 (c), it would be interesting to see the effect of the size of task-specific and feature-specific components on the performance. Or does the MK-MMD term incentivise the use of parameter free operators in the network as recently noted in Chen et al 2020 (https://arxiv.org/pdf/2002.05283.pdf)?
- Minor remark: I think the paper can benefit from one through proofread for grammatical corrections and notational consistencies. For instance, in the line above Section 4, x_{t} should be changed to x^{t} and \mathcal{L} in the line following eq(10) should include the superscript of “s”. Similarly, if the convention of underlined numbers in tables refers to second-best models that it should be made consistent across all tables.

---

### Decision · Program_Chairs · 2021-01-07
**Final Decision**

**Decision:**

Reject

**Comment:**

This work studies an intriguing problem of searching optimal architectures for unsupervised domain adaptation. It is based on a two-stage approach: (1) transferable architecture search via DARTS + MK-MMD; (2) transferable feature learning via Backbone + MCD.

The reviews for this paper are very insightful, constructive and of high quality. While all reviewers acknowledge the contribution of a new research problem, they unanimously have suggestions for further improving the paper:
- The novelty and soundness in the technical method are not fantastic. Authors should consider more elaborated loss designs for the approach, and unify the two stages with the same optimization objective.
- The empirical evaluation is by far not extensive and insightful. More stronger NAS approaches and larger-scale datasets should be included to give more evidence to support the claims that NAS-DA is better.
- A featured analysis of how non-iid architecture (as searched out by this work) differs from iid architecture would make the paper much more interesting.

Authors did not participate in the rebuttal and discussion phase.

AC scanned through the paper and believes that this paper studies a promising research direction, but the work cannot be accepted before addressing the reviewers' comments. The weaknesses are quite obvious and will have a high probability of being asked by the reviewers of the next conference. So the authors need to make sure that they substantially revise their work before submitting to yet another top venue.